# *LTBP3* Frameshift Variant in British Shorthair Cats with Complex Skeletal Dysplasia

**DOI:** 10.3390/genes12121923

**Published:** 2021-11-29

**Authors:** Gabriela Rudd Garces, Anna Knebel, Kirsten Hülskötter, Vidhya Jagannathan, Theresa Störk, Marion Hewicker-Trautwein, Tosso Leeb, Holger A. Volk

**Affiliations:** 1Institute of Genetics, Vetsuisse Faculty, University of Bern, 3001 Bern, Switzerland; gabriela.ruddgarces@vetsuisse.unibe.ch (G.R.G.); vidhya.jagannathan@vetsuisse.unibe.ch (V.J.); 2Institute of Veterinary Genetics “Ing. Fernando Noel Dulout”, National University of La Plata, La Plata 1900, Argentina; 3Department of Small Animal Medicine and Surgery, University of Veterinary Medicine Hannover, 30559 Hannover, Germany; anna.knebel@tiho-hannover.de (A.K.); holger.volk@tiho-hannover.de (H.A.V.); 4Department of Pathology, University of Veterinary Medicine Hannover, 30559 Hannover, Germany; kirsten.huelskoetter@tiho-hannover.de (K.H.); theresa.stoerk@tiho-hannover.de (T.S.); marion.hewicker-trautwein@tiho-hannover.de (M.H.-T.); 5Center for Systems Neuroscience Hannover (ZSN), 30559 Hannover, Germany

**Keywords:** *Felis catus*, skeletal dysplasia, bone, development, whole genome sequence, precision medicine, animal model

## Abstract

We investigated a highly inbred family of British Shorthair cats in which two offspring were affected by deteriorating paraparesis due to complex skeletal malformations. Radiographs of both affected kittens revealed vertebral deformations with marked stenosis of the vertebral canal from T11 to L3. Additionally, compression of the spinal cord, cerebellar herniation, coprostasis and hypogangliosis were found. The pedigree suggested monogenic autosomal recessive inheritance of the trait. We sequenced the genome of an affected kitten and compared the data to 62 control genomes. This search yielded 55 private protein-changing variants of which only one was located in a likely functional candidate gene, *LTBP3*, encoding latent transforming growth factor β binding protein 3. This variant, c.158delG or p.(Gly53Alafs*16), represents a 1 bp frameshift deletion predicted to truncate 95% of the open reading frame. LTBP3 is a known key regulator of transforming growth factor β (TGF-β) and is involved in bone morphogenesis and remodeling. Genotypes at the *LTBP3*:c.158delG variant perfectly co-segregated with the phenotype in the investigated family. The available experimental data together with current knowledge on *LTBP3* variants and their functional impact in human patients and mice suggest *LTBP3*:c.158delG as a candidate causative variant for the observed skeletal malformations in British Shorthair cats. To the best of our knowledge, this study represents the first report of *LTBP3*-related complex skeletal dysplasia in domestic animals.

## 1. Introduction

Skeletal dysplasias are a diverse group of phenotypes related to bone and cartilage development. Due to their heterogeneity, classification may be based on the genetic cause, lethality or involvement of similar skeletal parts [1]. Pathogenic variants affecting 437 different genes have been identified in human bone disorders [2].

In domestic animals, various skeletal dysplasias are known. In some instances, the selection for desired anatomical attributes during animal breeding has led to the genetic fixation of skeletal malformations, which have become breed standards. However, most skeletal dysplasias in domestic animals are considered diseases [3,4,5].

In cats, only a few skeletal dysplasias have been characterized at the molecular level. Osteochondrodysplasia in Scottish fold cats is caused by a missense variant in the *TRPV4* gene (000319-9685) [6,7]. A form of chondrodysplasia that represents the breed standard in short-legged Munchkin cats is caused by a large deletion within the *UGDH* gene encoding UDP-glucose 6-dehydrogenase (OMIA 000187-9685) [8,9]. The extreme brachycephaly in Burmese cats is at least partly due to a 12 bp deletion in the *ALX1* gene encoding the ALX homeobox 1 protein (OMIA 001551-9685) [10]. Polydactyly in cats is caused by non-coding SNVs in a regulatory region of the *SHH* gene encoding sonic hedgehog, a developmental morphogen (OMIA 000810-9685) [11]. A missense variant in the *ACVR1* gene encoding activin A receptor type I was reported in domestic shorthair cats with fibrodysplasia ossificans and secondary skeletal malformations (OMIA 000388-9685) [12].

There are several cat breeds, such as Pixie Bob, Kurilian Bobtail, American Bobcat, Japanese Bobcat and Manx, with shortened or kinked tails. Causative variants for these spinal malformations have been identified in the *TBXT* (OMIA 000975-9685) [13] and *HES7* genes (OMIA 001987-9685) [14,15].

The present study was initiated after a breeder reported two British Shorthair littermate kittens with severe skeletal deformations. The goal of this study was to characterize the clinical and radiological phenotype and to investigate a possible underlying causative genetic defect.

## 2. Materials and Methods

### 2.1. Animals

This study investigated two British Shorthair kittens affected by complex skeletal malformations. The affected cats, one male and one female, were littermates. We additionally obtained samples from both parents, six unrelated British Shorthair cats and 90 cats of other breeds.

### 2.2. Clinical Examinations

A general clinical and neurological examination was performed on the two affected kittens and their parents. Radiographs of the thorax and abdomen were obtained of both kittens in latero-lateral and ventro-dorsal projections. Post mortem computed tomography (CT, Philips Brilliance CT, Philips Medical Systems, Cleveland, OH, USA) and magnet resonance imaging (MRI, Phillips Achieva 3 Tesla MRI Scanner, Phillips Medical Systems) were performed on the euthanized male kitten (full body scan). Sagittal and transverse CT images and T1-weighted as well as T2-weighted images (MRI) were obtained. Cerebrospinal fluid (CSF) was taken post mortem by suboccipital puncture and analyzed.

### 2.3. Necropsy and Histopathological Examination

A full necropsy was performed of the male kitten after standard protocol. Tissue samples of all organs were taken and fixed in 10% neutral-buffered formalin for 24 h. Frozen tissue samples from the liver, pinna and skin were obtained for genetic analysis. Formalin fixed tissues were trimmed and embedded in paraffin, prior to routine histology, Luxol-Fast-Blue staining and immunohistochemistry on 2–3 µm thick tissue sections.

### 2.4. Immunohistochemistry

For immunohistochemistry, the tissue sections were deparaffinized and rehydrated by immersion in Roticlear (Carl Roth, A538.1, Karlsruhe, Germany) followed by isopropanol and 96% ethanol for 2 min each. Endogenous peroxidase was blocked with a solution of 85% ethanol with 0.5% H_2_O_2_ for 30 min at room temperature, prior to antigen retrieval by boiling for 20 min in 10 mM citrate buffer (pH 6) in the microwave. Unspecific bindings were blocked with goat serum (dilution 1:5) for 20 min. Primary antibodies (anti-Alzheimer precursor protein A4, a.a. 66–81 of APP (N-terminus), clone 22C11, Merck Millipore, Darmstadt, Germany; MAB348; dilution 1:2000; anti-pan-neuronal neurofilament marker mouse mAb SMI-311, Merck Millipore; NE1017, dilution 1:1000) were incubated at 4 °C overnight. The secondary antibody (biotinylated goat anti-mouse IgG, Vector BA9200; dilution 1:200) was incubated for 1.5 h, and positive signals were visualized using avidin–biotin–peroxidase complex (ABC Kit, Vectastain, PK6100, Vector Laboratories, Burlingame, CA, U.S.A.) and 0.1 g 3,3′-diaminobenzidine-tetrahydrochloride hydrate (DAB) in 200 mL PBS with 0.03% H_2_O_2_. The tissues were dehydrated in alcohol and mounted with Roti-Histokit II (Carl Roth, Karlsruhe, Germany, T160.1).

### 2.5. DNA Isolation

Genomic DNA was isolated from 500 μL EDTA blood samples of the cats in the study with the Maxwell RSC Whole Blood Kit, using a Maxwell RSC instrument (Promega, Dübendorf, Switzerland).

### 2.6. Whole Genome Sequencing of an Affected Kitten

An Illumina TruSeq PCR-free DNA library with ~400 bp insert size of an affected British Shorthair kitten was prepared. We collected 296 million 2 × 150 bp paired-end reads on a NovaSeq 6000 instrument (32.3× coverage). The reads were mapped to the Felis_catus_9.0 cat reference genome assembly as previously described [16]. The se-quence data were deposited under study accession PRJEB7401 and sample accession SAMEA8609185 at the European Nucleotide Archive.

### 2.7. Variant Calling

Variant calling was performed as described [16]. To predict the functional effects of the called variants, SnpEff [17] software together with NCBI annotation release 104 for the Felis_catus_9.0 genome reference assembly was used. For variant filtering, we used 62 control genomes (Appendix A). We employed a hard filtering approach to identify variants at which the affected cat was homozygous for the alternate allele (1/1) while 62 control genomes were either homozygous for the reference allele (0/0) or had a missing genotype call (./.). The output of the variant filtering is given in Appendix A.

### 2.8. Gene Analysis

We used the Felis_catus_9.0 cat reference genome assembly and NCBI annotation release 104. Numbering within the feline *LTBP3* gene corresponds to the NCBI RefSeq accession numbers XM_023240055.1 (mRNA) and XP_023095823.1 (protein).

### 2.9. Targeted Genotyping and Sanger Sequencing

The *LTBP3*:c.158delG variant was genotyped by direct Sanger sequencing of PCR amplicons. A 557 bp PCR product was amplified from genomic DNA using AmpliTaqGold360Mastermix (Thermo Fisher Scientific, Waltham, MA, U.S.A.) together with primers 5′-CGCTTCCTCTGTTCCTCTCC-3′ (Primer F) and 5′-AGACCCTACCCCACCAGGTA-3′ (Primer R). A 5200 Fragment Analyzer was used for the quality control of PCR products (Agilent, Santa Clara, CA, USA). After treatment with shrimp alkaline phosphatase and exonuclease I, PRC amplicons were sequenced on an ABI 3730 DNA Analyzer (Thermo Fisher Scientific). Sanger sequences were analyzed using the Sequencher 5.1 software (GeneCodes, Ann Arbor, MI, USA). The *CAPN1*:c.1295G>A variant was genotyped with the same methodology and these primers for the PCR amplification: 5′-CCTCCTACAGCCACCTTCTG-3′ (Primer F) and 5′- GTTCGTCGTGATCGTGTGAT-3′ (Primer R). Genotypes at the *CAPN1* and *LTBP3* variants were also extracted from the gvcf-file of the 99 Lives consortium (version with 340 cats) as described [8].

## 3. Results

### 3.1. Phenotype Description

Two British Shorthair littermates, one male and one female, with deteriorating paraparesis and their unaffected parents were investigated. The litter consisted of two affected and three non-affected kittens. The breeder noticed first signs of hind limb paraparesis in the affected kittens at 8 weeks of age. At 10 weeks of age, a clinical and neurological examination demonstrated lordosis and scoliosis, T3-L3 myelopathy and reduced motility of the intestine. Both kittens showed ambulatory paraparesis (Video S1). Hematology and cerebrospinal fluid analyses were inconspicuous. Radiography demonstrated severe vertebral column deformations and marked coprostasis (Figure 1 and Appendix A). Due to the severity of the clinical signs, the kittens were euthanized.

The post mortem examination of the male affected kitten by CT and MRI confirmed the deformation of multiple thoracic vertebral bodies. From T11 to L3, there was moderate to marked stenosis of the vertebral canal (lateral narrowing) as well as secondary compression of the spinal cord tissue, which could have led to the T3-L3 myelopathy. The MRI images also revealed an ascending and descending dilation of the central canal of the spinal cord (hydromyelia) and cerebellar herniation (Figure 2).

At necropsy, the multiple skeletal malformations with shortened legs, deviations of the spine and flattening of the occiput with narrowing of the caudal cranial fossa were corroborated. Parts of the caudal cerebellum and vermis were irreversibly dislocated into the foramen magnum with prominent indented deformation (Appendix A). The thoracic vertebral column showed a mild dorsal bend (kyphosis) with a following, moderate, ventral deformation (lordosis) and a minor lateral deviation (scoliosis) accompanied by a focal stenosis of the spinal canal at T11–12. The ventral cortical laminar bone showed an increased density and thickening up to 1 mm and the woven bone of the vertebral body was irregularly arranged (Appendix A). The ventral cortical laminar bone of the vertebral bodies showed an increased density and thickening up to 1 mm and the woven bone of the vertebral body and femur was irregularly arranged (Appendix A). Except for the compression of the caudal cerebellum, the histopathological examination of the brain was unremarkable. The compression of the thoracic spinal cord was associated with myelin damage accentuated in the dorso-lateral funiculi but also seen in the ventral aspects with dilation of myelin sheaths, axonal swelling (spheroid formation) and degeneration (Appendix A). The coprostasis was caused by annular constrictions of the colon and rectum with distention of anterior aspects (Appendix A). The constricted areas of the intestine showed a marked hypertrophy of the muscle layer (Appendix A) with loss of neurons in the submucosal (Meissner) and myenteric (Auerbach) plexus (hypoganglionosis). The thickening of the intestinal wall was accompanied by proliferation of vascularized fibrous connective tissue (granulation tissue) in the submucosa, between the muscle layers as well as the subserosa. Despite the loss of neurons, there was an excessive proliferation of nerve fibers interwoven into the granulation tissue and crossing the muscular layers (Appendix A).

### 3.2. Genetic Analysis

The two affected kittens came from a highly inbred family. The sire and dam were half siblings. The pedigree relationships were suggestive for a monogenic autosomal recessive mode of inheritance of the trait (Figure 3).

The genome of the male affected kitten was sequenced at 32.3× coverage and variants were called with respect to the Felis_catus_9.0 reference genome assembly. Subsequently, we searched for private homozygous variants in the genome of the affected cat that were not present in the 62 control genomes. The variant calling pipeline detected 55 private protein-changing variants in 28 genes (Table 1).

We then prioritized the 55 protein-changing variants according to the functional knowledge on the altered genes. Two closely linked variants on chromosome D1 affected genes with potential relevance for the observed phenotype. The affected cat had a missense variant in the *CAPN1* gene, XP_023095818.1:p.(Arg432His), and a frameshift deletion in the *LTBP3* gene encoding the latent transforming growth factor β binding protein 3 (Figure 4). The *LTBP3* variant was considered the most likely cause for the observed phenotype (see discussion). This candidate variant, a 1 bp deletion in the first exon of *LTBP3*, can be designated as ChrD1:110,690,432delC (Felis_catus_9.0 assembly). It is a frameshift variant, XM_023240055.1:c.158delG, predicted to truncate 95% of the open reading frame, XP_023095823.1:p.(Gly53Alafs*16).

We genotyped the *CAPN1* and *LTBP3* variants in the parents and the two affected littermates by Sanger sequencing. Both cases were homozygous for the mutant allele while the parents were heterozygous carriers. The segregation of the genotypes was compatible with a monogenic autosomal recessive mode of inheritance. We also genotyped the two variants in 6 additional unrelated British Shorthair cats and 90 other genetically diverse cats. The mutant *CAPN1* and *LTBP3* alleles were absent from all the tested unrelated control cats (Table 2).

We finally extracted the genotypes at these two variants from 340 cat genomes of the 99 Lives Consortium. A total of 8 random-bred cats of the 340 cats carried the mutant *CAPN1* allele in a heterozygous state (allele frequency 1.2%), while none of the cats carried the mutant *LTBP3* allele.

## 4. Discussion

The present article describes the clinical and pathological findings of a new form of inherited skeletal dysplasia in two full sibling British Shorthair cats.

The detected skull malformation with indentation of the occipital bone and cerebellar herniation through the foramen magnum resembles the “Chiari-like malformation” of dogs [18,19,20,21]. As the histopathological examination of the cerebellum and brain stem was unremarkable and there were no signs of syringomyelia, it is likely that the ataxia in the presented kittens was not solely caused by the protrusion of the cerebellar vermis into the foramen magnum. More likely, the reported ataxia and weakness of hind limbs were the consequence of the degenerative processes in the thoracic spinal cord and due to spinal cord compression.

The density and orientation of the trabecular bone was abnormal, which could indicate problems with the physiological remodeling. Nonetheless, it is unclear if a possible impairment of bone remodeling is fully responsible for the skeletal deformations or if the abnormal body posture and gait of the kittens may have contributed to the aberrant orientation of trabecular bone.

Histopathological examination of the colon revealed a segmental, distal hypoganglionosis with hypertrophy of the musculature and nerve fibers, as well as chronic inflammation with the formation of granulation tissue. These findings resemble another case report of a kitten with congenital colonic hypoganglionosis [22] and the diagnostic criteria for the congenital megacolon, known as “Hirschsprung’s disease” (HSCD) in humans [23]. The aganglionosis, seen in HSCD, is occasionally seen together with complex skeletal malformations [24] and could be caused by impaired migration and differentiation of neural stem cells from the neural crest [25]. However, in the studied cats, a primary neural crest defect seems unlikely. Our data suggest that the skeletal dysplasia caused secondary neuronal damage during pre- and postnatal development.

The genetic analysis revealed two tightly linked private protein-changing variants in the affected cat, a missense variant in *CAPN1* and a frameshift variant in *LTBP3*. *CAPN1* loss-of-function variants lead to autosomal recessive spastic paraplegia-76 in human patients [26] and spinocerebellar ataxia in Parson Russell Terriers [27]. It is unclear whether the *CAPN1*:p.Arg432His variant in the British Shorthair cat has any impact on calpain 1 function. Even if calpain 1 function in the affected cat was compromised, this would not explain the skeletal phenotype, but it may have possibly contributed to the neurological phenotype. As the mutant *CAPN1* allele was also present at a low frequency in the 99 Lives dataset, we consider a major functional impact of the *CAPN1* variant unlikely.

The *LTBP3*:c.158delG variant represents an excellent candidate causative variant for the skeletal phenotype. *LTBP3* encodes the latent transforming growth factor β binding protein 3, which is highly expressed in the heart, lungs, and bones and has a known role in TGF-β signaling and bone homeostasis [28,29].

TGF-β with its three isoforms encoded by the *TGFB1*, *TGFB2* and *TGFB3* genes is an important cytokine involved in the growth and remodeling of bones and many other tissues [30]. TGF-β is synthesized as an inactive precursor molecule that is proteolytically cleaved in the endoplasmic reticulum to yield a TGF-β homodimer in complex with the latency associated peptide (LAP). This complex is termed small latent complex (SLC). LTBP3 and other members of the LTBP family bind to the SLC, form covalent disulfide bonds with the LAP [31] and yield the large latent complex (LLC). The LLC is further processed in the Golgi apparatus, exported from the cell, and finally deposited in the extracellular matrix [32]. Activation of TGF-β signaling requires degradation of the LAP or release of active TGF-β homodimers from the LLC by other mechanisms [29,30,31]. LTBP3 and other members of the LTBP family are required to mediate the export and anchoring of the LLC to the extracellular matrix. LTBP3 deficiency reduces TGF-β activation and, therefore, may diminish associated cell proliferation and osteogenic differentiation [33]. Deregulation of the TGF-β signaling pathway is associated with numerous human diseases with prominent involvement of the skeletal system [34].

Human patients with bi-allelic loss-of-function variants in *LTBP3* develop dental anomalies and short stature (DASS, OMIM #601216). DASS is inherited as an autosomal recessive trait and clinical features may include oligodontia, amelogenesis imperfecta, short stature, brachyolmia and scoliosis, platyspondyly, cone-shaped epiphysis, undertubulation of the long bones, and in some instances also cardiac defects [35,36,37,38,39].

Other *LTBP3* variants cause an autosomal dominant phenotype in humans, termed geleophysic dysplasia 3 (GPHYSD3, OMIM #617809). The phenotype of GPHYSD3 overlaps with DASS and involves short stature, brachydactyly, restricted movements in the elbow and wrist joints, and dysmorphic facial features [40,41].

*Ltbp3^−/−^* homozygous null mice had shorter endochondral bones and smaller overall size compared to their wild-type littermates. *Ltbp3^−/−^* mice also developed craniofacial malformations and curvature of thoracic/cervical vertebrae (scoliosis). In addition, histological examination of *Ltbp3*^−/−^ skeletons revealed an increase in bone mass and persistence of cartilage remnants in trabecular bone, thus indicating a defect in bone resorption. A lack of Ltbp3 was suggested to result in decreased levels of TGF-β in bone and cartilage, which disrupts osteoclast function and bone turnover [42,43,44].

The *LTBP3* frameshift variant, c.158delG, p.(Gly53Alafs*16), in the British Shorthair cats of this study leads to a very early premature stop codon. We consider it, therefore, unlikely that any functional LTBP3 protein is expressed in homozygous mutant cats. Unfortunately, an experimental confirmation on the transcript or protein level could not be performed, as no suitable tissue samples were available. However, the genotype–phenotype co-segregation in the family together with the existing knowledge on the functional effect of *LTBP3* variants in humans and mice with skeletal dysplasia phenotypes support *LTBP3*:c.158delG as a candidate causative variant for the skeletal phenotype in the affected British Shorthair cats.

## 5. Conclusions

In summary, we characterized a new, recessively inherited form of skeletal dysplasia in British Shorthair cats and identified a frameshift variant in the *LTBP3* gene, c.158delG, as a candidate causative variant. Our data enable genetic testing to avoid the unintentional breeding of further affected cats and provide a potential spontaneous large animal model for *LTBP3*-related skeletal dysplasia.

## Figures and Tables

**Figure 1 genes-12-01923-f001:**
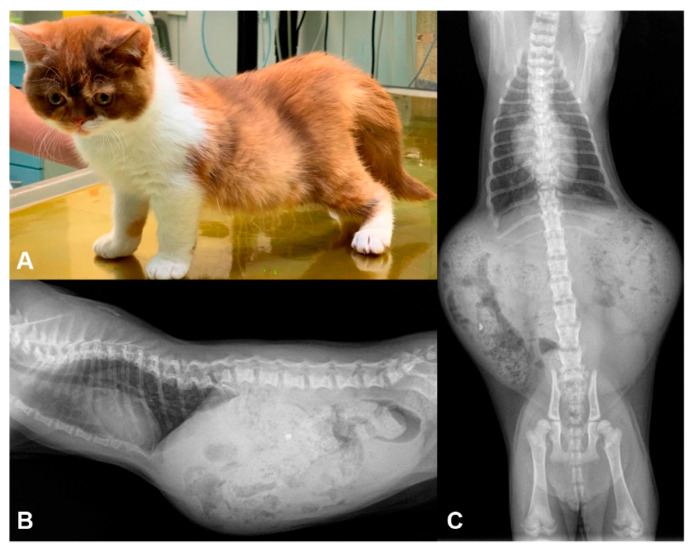
Clinical phenotype. (**A**) Photograph of the affected female kitten. (**B**) Radiograph in latero-lateral projection of the female kitten. (**C**) Radiograph in ventro-dorsal projection of the female kitten. Note the lordosis of this kitten, the malformation of the vertebral column within the thoracic region and the enlarged abdomen due to coprostasis.

**Figure 2 genes-12-01923-f002:**
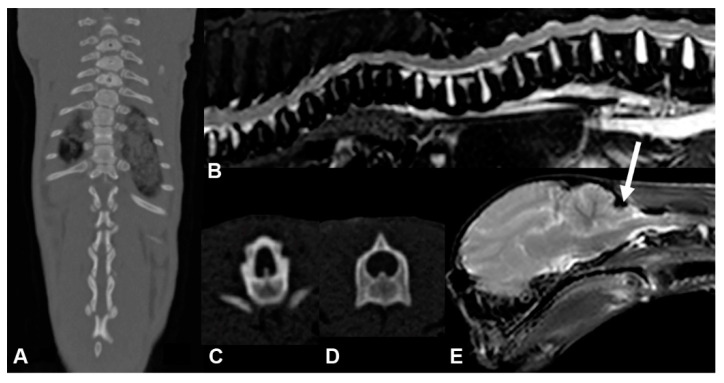
CT/MRI images of the affected male kitten. (**A**) CT image of the thoracic region in dorsal view. (**B**) T2-weighted sagittal MRI image of the vertebral column. (**C**) Transverse CT image of the vertebral body at the level of Th13. (**D**) Transverse CT image of the vertebral body at the level of L5. (**E**) T2-weighted sagittal MRI image of the brain with cerebellar herniation (white arrow).

**Figure 3 genes-12-01923-f003:**
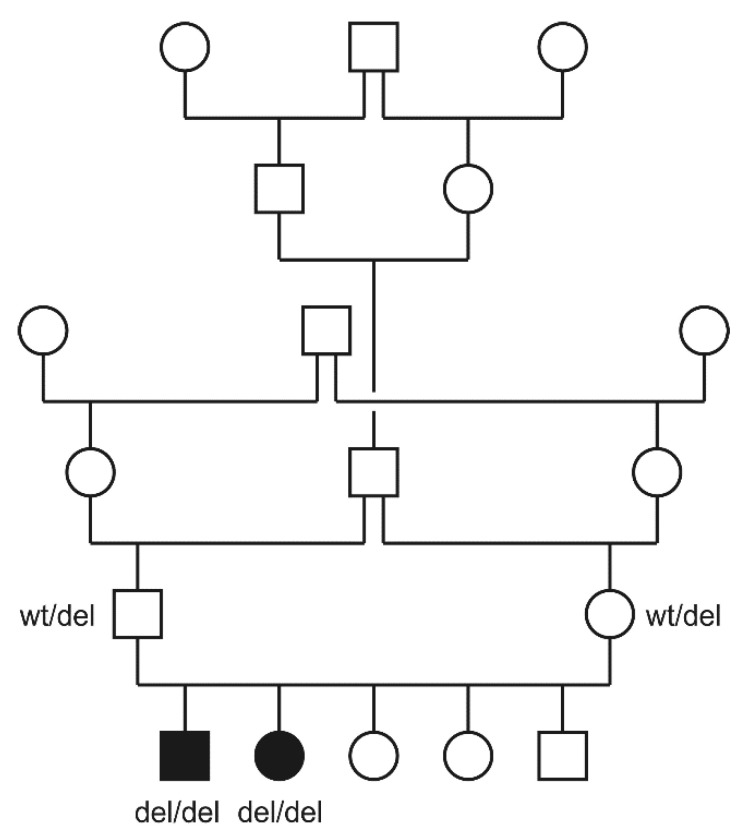
Pedigree of the investigated British Shorthair family. Filled symbols indicate affected cats and open symbols indicate non-affected cats. Squares and circles represent males and females, respectively. Note the multiple inbreeding loops in this pedigree. Genotypes at the *LTBP3*:c.158del variant are given for cats from which the samples were available.

**Figure 4 genes-12-01923-f004:**
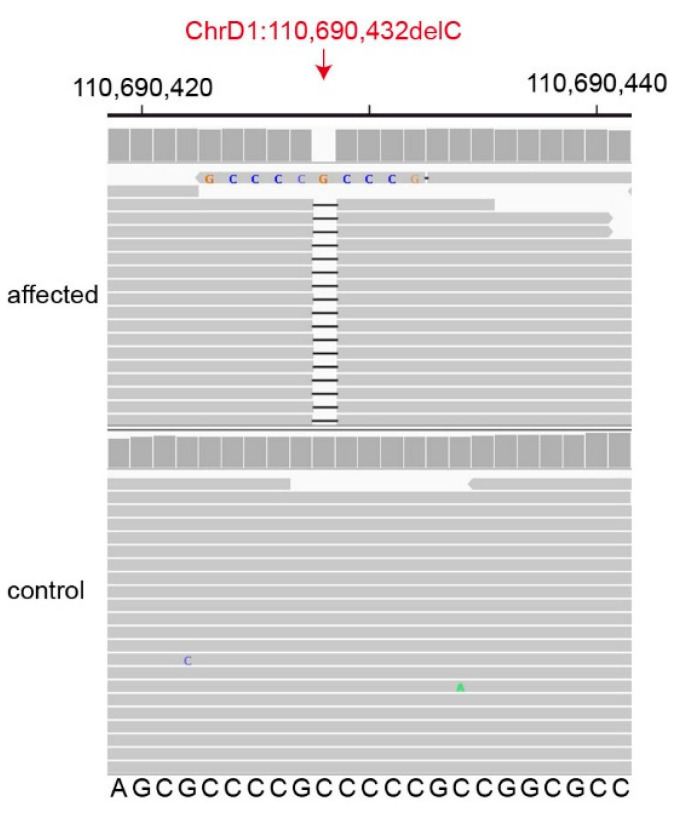
Details of the *LTBP3*:c.158delG, p.(Gly53Alafs*16) variant. Integrative Genomics Viewer (IGV) screenshot showing the short-read alignments of the affected kitten and a control cat at the position of the deletion, which affects the coding part of the first exon. Note that in the IGV screenshot, base 110,690,428 is deleted, which represents the first possible position of the deletion. Taking into consideration the 3′-rule of HGVS nomenclature, the correct variant designation is ChrD1:110,690,432delC (Felis_catus_9.0 assembly).

**Table 1 genes-12-01923-t001:** Homozygous variants detected by whole genome re-sequencing of an affected cat.

Filtering Step	Variants
Variants in whole genome	5,759,180
Private variants (absent from 62 control genomes)	7398
Protein-changing private variants	55

**Table 2 genes-12-01923-t002:** Association of the genotypes at the *CAPN1* and *LTBP3* variants with the skeletal malformation.

Phenotype	*CAPN1*:c.1295G>A	*LTBP3*:c.158delG
Cases (*n* = 2)	A/A	del/del
Non-affected parents (*n* = 2)	G/A	wt/del
Non-affected British Shorthair cats (*n* = 6)	G/G	wt/wt
Random-bred cats and cats from other breeds (*n* = 90)	G/G	wt/wt

## Data Availability

The genome sequence data used in this study are available from the European Nucleotide Archive. Accessions are given in Appendix A.

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
