# Peer review of "LTBP3 Frameshift Variant in British Shorthair Cats with Complex Skeletal Dysplasia"

_genes, 2021, doi:10.3390/genes12121923_

Round 1

Reviewer 1 Report

The investigators characterized this interesting defect segregating in these cats and systematically approached identification of the genetic locus responsible. While further studies will be needed to establish the mechanism by which LTBP3 affects skeletal development, the frameshift mutation that they have identified is an excellent candidate for this defect. The paper is well written, the analysis is doing and the supplementary material provides a detailed description of this defect which may be useful in the identification of similar abnormalities in cats and other species.

The authors have done an excellent job of characterizing this defect and identifying a candidate molecular lesion. 

Author Response

The investigators characterized this interesting defect segregating in these cats and systematically approached identification of the genetic locus responsible. While further studies will be needed to establish the mechanism by which LTBP3 affects skeletal development, the frameshift mutation that they have identified is an excellent candidate for this defect. The paper is well written, the analysis is doing and the supplementary material provides a detailed description of this defect which may be useful in the identification of similar abnormalities in cats and other species.

Response: Thank you very much for the compliments.

Reviewer 2 Report

Skeletal dysplasias are a group of complex bones/cartilage disorders observed in human and other animals. In this study, the authors recruited two siblings of British Shorthair (BSH) cats with complex skeletal malformations and launched a study of the phenotype and the genetics of skeletal dysplasia in cats.

First, the authors conducted clinical and histopathological examinations to characterize the skeleton malformation of these two affected cats. Marked stenosis of the vertebral canal from T11 to L3 and defects in neural system were identified in the affected cats.

Then, WGS was done in one affected BSH cat and, combined with 62 control cat genomes from public database, the authors aimed to identify the genetic cause of the cat skeletal dysplasia. They focused on the private variants that were homozygous in the affected cat while absent from the 62 controls. Fifty-five protein-changing private variants were identified, including a reading frame shift mutation p.G53Vfs*16 in LTBP3. Given the role of LTBP3 involved in the skeleton development of human and mice and the potential defect of LTBP3 caused by p.G53Vfs*16, the authors suggested p.G53Vfs*16 in LTBP3 as a strong candidate causal variant. The p.G53Vfs*16 of LTBP3 was validated in the affected BSH pedigree as well as other 436 cats from multiple origins.

I understand the reason why the authors considered p.G53Vfs*16 in LTBP3 as the strongest candidate causal mutation. However, I am not completely convinced that the other 54 protein-changing private variants could be confidently excluded. As Table S2 showed, besides p.G53Vfs*16 in LTBP3, there are 15 reading-frame-shift mutations in 7 genes (PLIN4, LOC111560563, LOC101096995, TNC, LOC111557471, LOC101088376, CRX), which might cause loss-of-function for the affected genes. We cannot neglect the possibility that one of the seven genes (especially the uncharacterized ones) may play a role in skeleton development, whose loss-of-function may cause skeleton malformation. In addition, non-synonymous mutations were identified in skeleton development genes (LGALS3, CLEC4A and CHPF), it is likely that one of these non-synonymous mutations might be important. I would suggest adding analysis to predict the impact of all the non-synonymous private variations on protein function, and test the co-segregation of the non-synonymous mutations with significant impact and reading frame shift mutations in the BSH pedigree. By doing this analysis, I believe most of the protein-changes private variations could be excluded. And the discussion should be addressed accordingly.

Several minor issues are:

  1. Line 64, Lyons et al (2016) BMC Genomics 17:265 should be cited.
  2. Line 106-108, be more specific about the sample using for DNA extraction.
  3. The region of LTBP3 first exon should be specified in Figure 4.
  4. Line 205, a separate table to summarize the protein-changing private variants, gene name, gene function, and the potential impact on protein function would be nice.

Author Response

(1)

I understand the reason why the authors considered p.G53Vfs*16 in LTBP3 as the strongest candidate causal mutation. However, I am not completely convinced that the other 54 protein-changing private variants could be confidently excluded. As Table S2 showed, besides p.G53Vfs*16 in LTBP3, there are 15 reading-frame-shift mutations in 7 genes (PLIN4, LOC111560563, LOC101096995, TNC, LOC111557471, LOC101088376, CRX), which might cause loss-of-function for the affected genes. We cannot neglect the possibility that one of the seven genes (especially the uncharacterized ones) may play a role in skeleton development, whose loss-of-function may cause skeleton malformation. In addition, non-synonymous mutations were identified in skeleton development genes (LGALS3, CLEC4A and CHPF), it is likely that one of these non-synonymous mutations might be important. I would suggest adding analysis to predict the impact of all the non-synonymous private variations on protein function, and test the co-segregation of the non-synonymous mutations with significant impact and reading frame shift mutations in the BSH pedigree. By doing this analysis, I believe most of the protein-changes private variations could be excluded. And the discussion should be addressed accordingly.

Response: We visually inspected the short-read alignments of all 18 variants with SnpEff predicted high impact (frame-shift & splice site variants) in IGV. While the LTBP3:c.158delG variant showed excellent support by the short-read alignments, all 17 other “private high impact variants” in Table S2 had clear indications of technical artifacts and incorrect variant calls or incorrect variant effect predictions. Most of the hypothetical LOC… gene predictions were recently retracted by NCBI in their newest genome annotation release 105. Other variant predictions were clearly due to errors in the FelCat 9 reference genome assembly. We added a column “Remark” to Table S2 and explained for each of the “private high impact variants” why we consider it incorrect.

We did the same visual inspection for the 36 variants with SnpEff predicted moderate impact (missense and in-frame indel variants). Here, 22 of the predicted moderate impact variants were not well supported (also explained in the revised Table S2).

The three missense variants in CHPF, CLEC4A and LGALS3, which were specifically proposed by the reviewer as potential candidate variants for a skeletal phenotype were all well supported by the experimental sequencing data. However, to the best of our knowledge, none of these genes has a confirmed role in skeletal development or skeleton homeostasis. Neither OMIM nor OMIA lists any known mutant phenotypes for any of these three genes. CLEC4A shows tissue-specific expression in cells of the immune system (proteinatlas.org). Lgals3-/- knockout mice do not show any overt phenotype with the exception of a mild immunological phenotype and slightly delayed wound healing (Hsu et al. 2000, Am J Pathol).

We agree with the reviewer that modifying effects of other variants on the specific phenotype of the investigated cats cannot be definitively excluded. However, in our opinion, the three specifically proposed missense variants are not likely to affect the skeletal phenotype. Furthermore, such hypothetical modifying effects would not be restricted to variants that are private to the affected cats. We think that our discussion of the CAPN3:p.Arg432His variant already reflects the limitations of the study and alerts the reader to potentially modifying effects of other genetic variants.

Unfortunately, the first author of the study, Gabriela Rudd Garces, has left our laboratory. We are therefore not able to perform additional genotyping experiments within a reasonable time frame and definitively not within the 3 days allowed for the minor revision.

(2)

Several minor issues are:

Line 64, Lyons et al (2016) BMC Genomics 17:265 should be cited.

Response: Yes absolutely. We added the lacking reference.

(3)

Line 106-108, be more specific about the sample using for DNA extraction.

Response: Revised accordingly.

(4)

The region of LTBP3 first exon should be specified in Figure 4.

Response: The coding part of the first exon spans positions 110,690,258-110,690,585 on chromosome D1 of the Felis_Catus_9.0 assembly. As we show only 21 internal nucleotides out of the 328 coding nucleotides in Figure 4, we don’t see an intuitive way to include this information in the figure. We slightly revised the figure legend to add this information.

(5)

Line 205, a separate table to summarize the protein-changing private variants, gene name, gene function, and the potential impact on protein function would be nice.

Response: We think that most of the requested information is already given in Table S2. A full listing would need more than one page, which would not look good in the main manuscript. “Gene function” is very difficult to summarize in a meaningful way in a table. There are many genes whose functions are partially or completely unknown. A major fraction of the gene annotations changed from annotation release 104 to 105 a few days ago. Including the requested table would also mean to include a large amount of outdated information. On the other hand, as we are confident to having identified the true causal variant, we see no point in repeating our entire analysis with respect to the new genome assembly and annotation, which would require several months of time.